# The Information Loss of a Stochastic Map

**DOI:** 10.3390/e23081021

**Published:** 2021-08-08

**Authors:** James Fullwood, Arthur J. Parzygnat

**Affiliations:** 1School of Mathematical Sciences, Shanghai Jiao Tong University, 800 Dongchuan Road, Shanghai 200240, China; 2Institut des Hautes Études Scientifiques, 35 Route de Chartres, 91440 Bures-sur-Yvette, France

**Keywords:** Bayes, conditional probability, disintegration, entropy, error correction, functor, information theory, Markov category, stochastic map, synthetic probability, Primary 94A17, Secondary 18A05, 62F15

## Abstract

We provide a stochastic extension of the Baez–Fritz–Leinster characterization of the Shannon information loss associated with a measure-preserving function. This recovers the conditional entropy and a closely related information-theoretic measure that we call *conditional information loss*. Although not functorial, these information measures are semi-functorial, a concept we introduce that is definable in any Markov category. We also introduce the notion of an *entropic Bayes’ rule* for information measures, and we provide a characterization of conditional entropy in terms of this rule.

## 1. Introduction

The information loss K(f) associated with a measure-preserving function $\left(X,p\right)\overset{f}{\to }\left(Y,q\right)$ between finite probability spaces is given by the Shannon entropy difference
K(f):=H(p)−H(q),
where $H\left(p\right):=-\sum_{x\in X}p_{x}\log p_{x}$ is the Shannon entropy of *p* (and similarly for *q*). In [1], Baez, Fritz, and Leinster proved that the information loss satisfies, and is uniquely characterized up to a non-negative multiplicative factor by, the following conditions:0.Positivity: K(f)≥0 for all $\left(X,p\right)\overset{f}{\to }\left(Y,q\right)$. This says that the information loss associated with a deterministic process is always non-negative.1.Functoriality: K(g∘f)=K(g)+K(f) for every composable pair (f,g) of measure-preserving maps. This says that the information loss of two successive processes is the sum of the information losses associated with each process.2.Convex Linearity: K(λf⊕(1−λ)g)=λK(f)+(1−λ)K(g) for all λ∈(0,1). This says that the information loss associated with tossing a (possibly unfair) coin in deciding amongst two processes is the associated weighted sum of their information losses.3.Continuity: K(f) is a continuous function of *f*. This says that the information loss does not change much under small perturbations (i.e., is robust with respect to errors).

As measure-preserving functions may be viewed as deterministic stochastic maps, it is natural to ask whether there exist extensions of the Baez–Fritz–Leinster (BFL) characterization of information loss to maps that are inherently random (i.e., stochastic) in nature. In particular, what information-theoretic quantity captures such an information loss in this larger category?

This question is answered in the present work. Namely, we extend the BFL characterization theorem, which is valid on *deterministic* maps, to the larger category of *stochastic* maps. In doing so, we also find a characterization of the conditional entropy. Although the resulting extension is not functorial on the larger category of stochastic maps, we formalize a weakening of functoriality that restricts to functoriality on deterministic maps. This weaker notion of functoriality is definable in any Markov category [2,3], and it provides a key axiom in our characterization.

To explain how we arrive at our characterization, let us first recall the definition of stochastic maps between finite probability spaces, for which the measure-preserving functions are a special case. A ***stochastic map*** $\left(X,p\right)\overset{f}{⇝}\left(Y,q\right)$ associates with every x∈X a probability distribution $f_{x}$ on *Y* such that $q_{y}=\sum_{x\in X}f_{yx}p_{x}$, where $f_{yx}$ is the distribution $f_{x}$ evaluated at y∈Y. In terms of information flow, the space (X,p) may be thought of as a probability distribution on the set of inputs for a communication channel described by the stochastic matrix $f_{yx}$, while (Y,q) is then thought of as the induced distribution on the set of outputs of the channel.

Extending the information loss functor by assigning H(p)−H(q) to any *stochastic* map $\left(X,p\right)\overset{f}{⇝}\left(Y,q\right)$ would indeed result in an assignment that satisfies conditions 1–3 listed above. However, it would no longer be positive and the interpretation as an information loss would be gone. Furthermore, no additional information about the stochasticity of the map *f* would be used in determining this assignment. In order to guarantee positivity, an additional term, depending on the stochasticity of *f*, is needed. This term is provided by the ***conditional entropy*** of $\left(X,p\right)\overset{f}{⇝}\left(Y,q\right)$ and is given by the the non-negative real number
$$H\left(f|p\right):=\sum_{x\in X}p_{x}H\left(f_{x}\right),$$
where $H(f_{x})$ is the Shannon entropy of the distribution $f_{x}$ on *Y* (in the case that (X,p) and (Y,q) are probability spaces associated with the alphabets of random variables X and Y, then H(f|p) coincides with conditional entropy H(Y|X) [4]). If $\left(X,p\right)\overset{f}{⇝}\left(Y,p\right)$ is in fact deterministic, i.e., if $f_{x}$ is a point-mass distribution for all x∈X, then $H\left(f_{x}\right)=H\left(f|p\right)=0$ for all x∈X. As such, H(f|p) is a measure of the uncertainty (or randomness) of the outputs of *f* averaged over the prior distribution *p* on the set *X* of its inputs. Indeed, H(f|p) is maximized precisely when $f_{x}$ is the uniform distribution on *Y* for all x∈X.

Therefore, given a stochastic map $\left(X,p\right)\overset{f}{⇝}\left(Y,q\right)$, we call
K(f):=H(p)−H(q)+H(f|p)
the ***conditional information loss*** of $\left(X,p\right)\overset{\;f}{⇝}\left(Y,q\right)$ (the same letter *K* is used here because it agrees with the Shannon entropy difference when *f* is deterministic). As H(f|p)=0 whenever *f* is deterministic, the conditional information loss restricts to the category of measure preserving functions as the information loss functor of Baez, Fritz, and Leinster, while also satisfying conditions 0, 2, and 3 (i.e., positivity, convex linearity, and continuity) on the larger category of stochastic maps. However, conditional information loss is *not* functorial in general, and while this may seem like a defect at first glance, we prove that there is no extension of the information loss functor that remains functorial on the larger category of stochastic maps if the positivity axiom is to be preserved, thus retaining an interpretation as information loss. In spite of this, conditional information loss does satisfy a weakened form of functoriality, which we briefly describe now.

A pair $\left(X,p\right)\overset{f}{⇝}\left(Y,q\right)\overset{g}{⇝}\left(Z,r\right)$ of composable stochastic maps is ***a.e. coalescable*** if and only if for every pair of elements z∈Z and x∈X for which $r_{z}>0$ and $p_{x}>0$, there exists a unique y∈Y such that $f_{yx}>0$ and $g_{zy}>0$. Intuitively, this says that the information about the intermediate step can be recovered given knowledge about the input and output. In particular, if *f* is deterministic, then the pair (f,g) is a.e. colescable (for obvious reasons, since knowing *x* alone is enough to determine the intermediate value). However, there are other many situations where a pair could be a.e. coalescable and the maps need not be deterministic. With this definition in place (which we also generalize to the setting of arbitrary Markov categories), we replace functoriality with the following weaker condition.

1^🟉^.Semi-functoriality: K(g∘f)=K(g)+K(f) for every a.e. coalescable pair $\left(X,p\right)\overset{f}{⇝}\left(Y,q\right)\overset{g}{⇝}\left(Z,r\right)$ of stochastic maps. This says that the conditional information loss of two successive processes is the sum of the conditional information losses associated with each process *provided* that the information in the intermediate step can always be recovered.

Replacing functoriality with semi-functoriality is not enough to characterize the conditional information loss. However, it comes quite close, as only one more axiom is needed. Assuming positivity, semi-functoriality, convex linearity, and continuity, there are several equivalent axioms that may be stipulated to characterize the conditional information loss. To explain the first option, we introduce a convenient factorization of every stochastic map $\left(X,p\right)\overset{f}{⇝}\left(Y,q\right)$. The ***bloom-shriek factorization*** of *f* is given by the decomposition $f=\pi_{Y}\circ {¡}_{f}$, where ${¡}_{f}:X⇝X\times Y$ is the ***bloom*** of *f* whose value at $x^{\prime }$ is the probability measure on X×Y given by sending (x,y) to $\delta_{x^{\prime }x}f_{yx^{\prime }}$, where $\delta_{x^{\prime }x}$ is the Kronecker delta. In other words, ${¡}_{f}$ records each of the probability measures $f_{x}$ on a copy of *Y* indexed by x∈X. A visualization of the bloom of *f* is given in Figure 1a. When one is given the additional data of probability measures *p* and *q* on *X* and *Y*, respectively, then Figure 1b illustrates the bloom-shriek factorization of *f*. From this point of view, ${¡}_{f}$ keeps track of the information encoded in *both*
*p* and *f*, while the projection map $\pi_{Y}$ forgets, or loses, *some* of this information.

With this in mind, our final axiom to characterize the conditional information loss is

4(a).Reduction: $K\left(f\right)=K\left(\pi_{Y}\right)$, where $f=\pi_{Y}\circ {¡}_{f}$ is the bloom-shriek factorization of *f*. This says that the conditional information loss of *f* equals the information loss of the projection using the associated joint distribution on X×Y.

Note that this axiom describes how *K* is determined by its action on an associated class of deterministic morphisms. These slightly modified axioms, namely, semi-functoriality, convex linearity, continuity, and reduction, characterize the conditional information loss and therefore extend Baez, Fritz, and Leinster’s characterization of information loss. A much simpler axiom that may be invoked in place of the reduction axiom which also characterizes conditional information loss is the following.

4(b).Blooming: $K({¡}_{p})=0$, where ${¡}_{p}$ is the unique map (•,1)⇝(X,p) from a one point probability space to (X,p). This says that if a process begins with no prior information, then there is no information to be lost in the process.

The conditional entropy itself can be extracted from the conditional information loss by a process known as Bayesian inversion, which we now briefly recall. Given a stochastic map $\left(X,p\right)\overset{f}{⇝}\left(Y,q\right)$, there exists a stochastic map $\left(Y,q\right)\overset{\bar{f}}{⇝}\left(Y,q\right)$ such that $f_{yx}p_{x}={\bar{f}}_{xy}q_{y}$ for all x∈X and y∈Y (the stochastic map $\bar{f}$ is the almost everywhere unique conditional probability so that Bayes’ rule holds). Such a map is called a ***Bayesian inverse*** of *f*. The Bayesian inverse can be visualized using the bloom-shriek factorization because it itself has a bloom-shriek factorization $\bar{f}=\pi_{X}\circ {¡}_{\bar{f}}$. This is obtained by finding the stochastic maps in the opposite direction of the arrows so that they reproduce the appropriate volumes of the water droplets.

Given this perspective on Bayesian inversion, we prove that the *conditional entropy* of $\left(X,p\right)\overset{f}{⇝}\left(Y,q\right)$ equals the *conditional information loss* of its Bayesian inverse $\left(Y,q\right)\overset{\bar{f}}{⇝}\left(X,p\right)$. Moreover, since the conditional information loss of $\bar{f}$ is just the information loss of $\pi_{X}$, this indicates how the conditional entropy and conditional information losses are the ordinary information losses associated with the two projections $\pi_{X}$ and $\pi_{Y}$ in Figure 1b. This duality also provides an interesting perspective on conditional entropy and its characterization. Indeed, using Bayesian inversion, we also characterize the *conditional entropy* as the unique assignment *F* sending measure-preserving stochastic maps between finite probability spaces to real numbers satisfying conditions 0, $1^{🟉}$, 2, and 3 above, but with a new axiom that reads as follows.

4(c).Entropic Bayes’ Rule: $F\left(f\right)+F\left({¡}_{p}\right)=F\left(\bar{f}\right)+F\left({¡}_{q}\right)$ for all $\left(X,p\right)\overset{f}{⇝}\left(Y,q\right)$. This is an information theoretic analogue of Bayes’ rule, which reads $f_{yx}p_{x}={\bar{f}}_{xy}q_{y}$ for all x∈X and y∈Y, or in more traditional probabilistic notation P(y|x)P(x)=P(x|y)P(y).

In other words, we obtain a *Bayesian* characterization of the conditional entropy. This provides an entropic and information-theoretic description of Bayes’ rule from the Markov category perspective, in a way that we interpret as answering an open question of Fritz [6].

## 2. Categories of Stochastic Maps

In the first few sections, we define all the concepts involved in proving that the conditional information loss satisfies the properties that we will later prove characterize it. This section introduces the domain category and its convex structure.

**Definition** **1.***Let X and Y be finite sets. A***stochastic map***f:X⇝Y associates a probability measure $f_{x}$ to every x∈X. If f:X⇝Y is such that $f_{x}$ is a point-mass distribution for every x∈X, then f is said to be***deterministic**.

**Notation** **1.**
*Given a stochastic map f:X⇝Y (also written as $X\overset{f}{⇝}Y$), the value $f_{x}\left(y\right)\in \left[0,1\right]$ will be denoted by $f_{yx}$. As there exists a canonical bijection between deterministic maps of the form X⇝Y and functions X→Y, deterministic maps from X to Y will be denoted by the functional notation X→Y.*


**Definition** **2.**
*A stochastic map of the form •⇝X from a single element set to a finite set X is a single probability measure on X. Its unique value at x will be denoted by $p_{x}$ for all x∈X. The set $N_{p}:=\left\{x\in X\;|\;p_{x}=0\right\}$ will be referred to as the*
**nullspace**
*of p.*


**Definition** **3.***Let***FinStoch***be the category of stochastic maps between finite sets. Given a finite set X, the identity map of X in***FinStoch***corresponds to the identity* function ${\mathrm{id}}_{X}:X\to X$. *Second, given stochastic maps f:X⇝Y and g:Y⇝Z, the composite g∘f:X⇝Z is given by the Chapmann–Kolmogorov equation ${\left(g\circ f\right)}_{zx}:=\sum_{y\in Y}g_{zy}f_{yx}.$*

**Definition** **4.***Let X be a finite set. The***copy***of X is the diagonal embedding $\Delta_{X}:X\to X\times X$, and the***discard***of X is the unique map from X to the terminal object • in***FinStoch**, *which will be denoted by ${!}_{X}:X\to •$. If Y is another finite set, the***swap map***is the map γ:X×Y→Y×X given by (x,y)↦(y,x). Given morphisms $f:X⇝X^{\prime }$ and $g:Y⇝Y^{\prime }$ in***FinStoch**, *the***product***of f and g is the stochastic map $f\times g:X\times Y⇝X^{\prime }\times Y^{\prime }$ given by ${\left(f\times g\right)}_{\left(x^{\prime },y^{\prime }\right)\left(x,y\right)}:=f_{x^{\prime }x}g_{y^{\prime }y}.$*

The product of stochastic maps endows **FinStoch** with the structure of a monoidal category. Together with the copy, discard, and swap maps, **FinStoch** is a Markov category [2,3].

**Definition** **5.**
*Let FinPS (this stands for “*
***fin***
*ite*
***p***
*robabilities and*
***s***
*tochastic maps”) be the co-slice category •↓FinStoch, i.e., the category whose objects are pairs (X,p) consisting of a finite set X equipped with a probability measure p, and a morphism from (X,p) to (Y,q) is a stochastic map $X\overset{f}{⇝}Y$ such that $q_{y}=\sum_{x\in X}f_{yx}p_{x}$ for all y∈Y. The subcategory of deterministic maps in FinPS will then be denoted by FinPD (which stands for “*
***fin***
*ite*
***p***
*robabilities and*
***d***
*eterministic maps”). A pair (f,g) of morphisms in FinPS is said to be a*
**composable pair**
*iff g∘f exists.*


Note that the category FinPD was called FinProb in [1].

**Remark** **1.**
*Though it is often the case that we will denote a morphism $\left(X,p\right)\overset{f}{⇝}\left(Y,q\right)$ in FinPS simply by f, such notation is potentially ambiguous, as the morphism $\left(X,p^{\prime }\right)\overset{f}{⇝}\left(Y,q^{\prime }\right)$ is distinct from the morphsim $\left(X,p\right)\overset{f}{⇝}\left(Y,q\right)$ whenever $p\neq p^{\prime }$. As such, we will only employ the shorthand of denoting a morphism in FinPS by its underlying stochastic map whenever the source and target of the morphism are clear from the context.*


**Lemma** **1.**
*The object (•,1) given by a single element set equipped with the unique probability measure is a zero object (i.e., terminal and initial) in FinPS.*


**Definition** **6.**
*Given an object (X,p) in FinPS, the*
**shriek**
*and*
**bloom**
*of p are the unique maps to and from (•,1) respectively, which will be denoted ${!}_{p}:\left(X,p\right)\to \left(•,1\right)$ and ${¡}_{p}:\left(•,1\right)⇝\left(X,p\right)$ (the former is deterministic, while the latter is stochastic). The underlying stochastic maps associated with ${!}_{p}$ and ${¡}_{p}$ are ${!}_{X}:X\to •$ and p:•⇝X, respectively.*


**Example** **1.**
*Since (•,1) is a zero object, given any two objects (X,p) and (Y,q), there exists at least one morphism (Y,q)⇝(X,p), namely the composite $\left(Y,q\right)\overset{{!}_{q}}{\to }\left(•,1\right)\overset{{¡}_{p}}{⇝}\left(X,p\right)$.*


**Definition** **7.**
*Let $\left(X,p\right)\overset{f}{⇝}\left(Y,q\right)$ be a morphism in FinPS. The*
**joint distribution**
*associated with f is the probability measure ϑ(f):•⇝X×Y given by $\vartheta {\left(f\right)}_{\left(x,y\right)}=f_{yx}p_{x}$.*


It is possible to take convex combinations of both objects and morphisms in FinPS, and such assignments will play a role in our characterization of conditional entropy.

**Definition** **8.**
*Let p:•⇝X be a probability measure and let ${\left\{\left(Y_{x},q^{x}\right)\right\}}_{x\in X}$ be a collection of objects in FinPS indexed by X. The p-*
**weighted convex sum**
*${⨁}_{x\in X}p_{x}\left(Y_{x},q^{x}\right)$ is defined to be the set ${∐}_{x\in X}Y_{x}$ equipped with the probability measure ${⨁}_{x\in X}p_{x}q^{x}$ given by*
$${\left(\underset{x\in X}{⨁}p_{x}q^{x}\right)}_{z}:=\left\{\begin{array}{c} p_{x}q_{z}^{x}\;\;if\;z\in \;Y_{x}\\ 0\;\;\;\mathrm{otherwise}.\; \end{array}\right.$$
*In addition, if $\left(Y_{x},q^{x}\right)\overset{Q^{x}}{⇝}\left(Y_{x}^{\prime },q^{\prime x}\right)$ is a collection of morphisms in FinPS indexed by X, the p-*
**weighted convex sum**
*${⨁}_{x\in X}p_{x}Q^{x}:\left({∐}_{x\in X}Y_{x},{⨁}_{x\in X}p_{x}q^{x}\right)⇝\left({∐}_{x\in X}Y_{x}^{\prime },{⨁}_{x\in X}p_{x}q^{\prime x}\right)$ is given by*
$${\left(\underset{x\in X}{⨁}p_{x}Q^{x}\right)}_{z^{\prime }z}:=\left\{\begin{array}{c} p_{x}{\left(Q^{x}\right)}_{z^{\prime }z}\;\;\mathrm{if}\;\left(z^{\prime },z\right)\in Y_{x}^{\prime }\times \;Y_{x}\\ 0\;\;\;\mathrm{otherwise}.\; \end{array}\right.$$


## 3. The Baez–Fritz–Leinster Characterization of Information Loss

In [1], Baez, Fritz, and Leinster (BFL) characterized the Shannon entropy difference associated with measure-preserving functions between finite probability spaces as the only non-vanishing, continuous, convex linear functor from FinPD to the non-negative reals (up to a multiplicative constant). It is then natural to ask whether there exist either extensions or analogues of their result by including non-deterministic morphisms from the larger category FinPS. Before delving deeper into such inquiry, we first recall in detail the characterization theorem of BFL.

**Definition** **9.**
*Let BR be the convex category consisting of a single object and whose set of morphisms is R. The composition in BR is given by addition. Convex combinations of morphisms are given by ordinary convex combinations of numbers. The subcategory of non-negative reals will be denoted $BR_{\geq 0}$.*


In the rest of the paper, we will not necessarily assume that assignments from one category to another are functors. Nevertheless, we do assume they form (class) functions (see ([7], Section I.7) for more details). Furthermore, we assume that they respect or reflect source and targets in the following sense. If C and D are two categories, all functions F:C→D are either ***covariant*** or ***contravariant*** in the sense that for any morphism $a\overset{\gamma }{\to }b$ in C, F(γ) is a morphism from F(a) to F(b) or from F(b) to F(a), respectively. These are the only types of functions between categories we will consider in this work. As such, we therefore abuse terminology and use the term *functions* for such assignments throughout. If *M* is a *commutative* monoid and BM denotes its one object category, then every covariant function C→BM is also contravariant and vice-versa.

We now define a notion of continuity for functions of the form F:FinPS→BR.

**Definition** **10.**
*A sequence of morphisms $\left(X_{n},p_{n}\right)\overset{f_{n}}{⇝}(Y_{n},q_{n})$ in FinPS*
**converges**
*to a morphism $\left(X,p\right)\overset{f}{⇝}\left(Y,q\right)$ if and only if the following two conditions hold.*
*(a)* 
*There exists an N∈N for which $X_{n}=X$ and $Y_{n}=Y$ for all n≥N.*
*(b)* 
*The following limits hold: $\lim_{n\to \infty }p_{n}=p$ and $\lim_{n\to \infty }f_{n}=f$ (note that these limits necessarily imply $\lim_{n\to \infty }q_{n}=q$).*

*A function F:FinPS→BR is*
**continuous**
*if and only if $\lim_{n\to \infty }F\left(f_{n}\right)=F\left(f\right)$ whenever $\left\{f_{n}\right\}$ is a sequence in FinPS converging to f.*


**Remark** **2.***In the subcategory FinPD, since the topology of the collection of* functions *from a finite set X to another finite set Y is* discrete*, one can equivalently assume that a sequence $f_{n}$ as in Definition 10, but this time with all $f_{n}$ deterministic, converges to $\left(X,p\right)\overset{f}{\to }\left(Y,q\right)$ if and only if the following two conditions hold.*
*(a)* *There exists an N∈N for which $X_{n}=X,Y_{n}=Y,$$f_{n}=f$ for all n≥N.**(b)* *For n≥N, one has $\lim_{n\to \infty }p_{n}=p$.*
*In this way, our definition of convergence agrees with the definition of convergence of BFL on the subcategory FinPD [1].*

**Definition** **11.**
*A function F:FinPS→BR is said to be*
**convex linear**
*if and only if for all objects (X,p) in FinPS,*
$$F\left(\underset{x\in X}{⨁}p_{x}Q^{x}\right)=\sum_{x\in X}p_{x}F\left(Q^{x}\right)$$
*for all collections $\left\{\left(Y_{x},q^{x}\right)\overset{Q^{x}}{⇝}(Y_{x}^{\prime },q^{\prime x})\right\}{}_{x\in X}$ in FinPS.*


**Definition** **12.**
*A function F:FinPS→BR is said to be*
**functorial**
*if and only if it is in fact a functor, i.e., if and only if F(g∘f)=F(f)+F(g) for every composable pair (f,g) in FinPS.*


**Definition** **13.**
*Let p:•⇝X be a probability measure. The*
**Shannon entropy**
*of p is given by*
$$H\left(p\right):=-\sum_{x\in X}p_{x}\log\left(p_{x}\right).$$


When considering any entropic quantity, we will always adhere to the convention that 0log(0)=0.

**Definition** **14.**
*Given a map $\left(X,p\right)\overset{f}{\to }\left(Y,p\right)$ in FinPD, the Shannon entropy difference H(p)−H(q) will be referred to as the*
**information loss**
*of f. Information loss defines a functor K:FinPD→BR, henceforth referred to as the*
**information loss functor**
*on FinPD.*


**Theorem** **1**(Baez–Fritz–Leinster [1])**.**
*Suppose $F:\mathrm{FinPD}\to BR_{\geq 0}$ is a function which satisfies the following conditions.*
*1.* *F is functorial.**2.* *F is convex linear.**3.* *F is continuous.*
*Then F is a non-negative multiple of information loss. Conversely, the information loss functor is non-negative and satisfies conditions 1–3.*

In light of Theorem 1, it is natural to question whether or not there exists a functor $K:\mathrm{FinPS}\to BR_{\geq 0}$ that restricts to FinPD as the information loss functor. It turns out that no such non-vanishing functor exists, as we prove in the following proposition.

**Proposition** **1.**
*If $F:\mathrm{FinPS}\to BR_{\geq 0}$ is a functor, then F(f)=0 for all morphisms f in FinPS.*


**Proof.** Let $\left(X,p\right)\overset{f}{⇝}\left(Y,q\right)$ be a morphism in FinPS. Since *F* is a functor,
$$F\left({¡}_{q}\right)=F\left(f\circ {¡}_{p}\right)=F\left(f\right)+F\left({¡}_{p}\right)\Rightarrow 0\leq F\left(f\right)=F\left({¡}_{q}\right)-F\left({¡}_{p}\right).$$
Let $\left(Y,q\right)\overset{g}{⇝}\left(X,p\right)$ be *any* morphism in FinPS (which necessarily exists by Example 1, for instance). Then a similar calculation yields
$$0\leq F\left(g\right)=F\left({¡}_{p}\right)-F\left({¡}_{q}\right)=-F\left(f\right).$$
Hence, F(f)=0. □

## 4. Extending the Information Loss Functor

Proposition 1 shows it is not possible to extend the information loss functor to a functor on FinPS. Nevertheless, in this section, we define a non-vanishing *function*$K:\mathrm{FinPS}\to BR_{\geq 0}$ that restricts to the information loss *functor* on FinPD, which we refer to as *conditional information loss*. While *K* is not functorial, we show that it satisfies many important properties such as continuity, convex linearity, and invariance with respect to compositions with isomorphisms. Furthermore, in Section 5 we show *K* is functorial on a restricted class of composable pairs of morphisms (cf. Definition 18), which are definable in any Markov category. At the end of this section we characterize conditional information loss as the unique extension of the information loss functor satisfying the reduction axiom 4(a) as stated in the introduction. In Section 8, we prove an intrinsic characterization theorem for *K* without reference to the deterministic subcategory FinPD inside FinPS. Appendix A provides an interpretation of the vanishing of conditional information loss in terms of correctable codes.

**Definition** **15.**
*The*
**conditional information loss**
*of a morphism $\left(X,p\right)\overset{f}{⇝}\left(Y,q\right)$ in FinPS is the real number given by*
K(f):=H(p)−H(q)+H(f|p),
*where*
$$H\left(f|p\right):=\sum_{x\in X}p_{x}H\left(f_{x}\right)$$
*is the*
**conditional entropy**
*of $\left(X,p\right)\overset{f}{⇝}\left(Y,q\right)$.*


**Proposition** **2.**
*The function K:FinPS→BR, uniquely determined on morphisms by sending $\left(X,p\right)\overset{f}{⇝}\left(Y,q\right)$ to K(f), satisfies the following conditions.*
*(i)* 
*K(f)≥0.*
*(ii)* 
*K restricted to FinPD agrees with the information loss functor (cf. Definition 14).*
*(iii)* 
*K is convex linear.*
*(iv)* 
*K is continuous.*
*(v)* 
*Given $\left(X,p\right)\overset{f}{⇝}\left(Y,q\right)$, then $K\left(f\right)=K\left(\pi_{Y}\right)$, where $\left(X\times Y,\vartheta \left(f\right)\right)\overset{\pi_{Y}}{\to }\left(Y,q\right)$ is the projection and ϑ(f) is the joint distribution (cf. Definition 7).*



**Lemma** **2.**
*Let $\left(X,p\right)\overset{f}{⇝}\left(Y,q\right)$ be a morphism in FinPS. Then*
$$K\left(f\right)=-\sum_{x\in X\backslash N_{p}}\sum_{y\in Y\backslash N_{f_{x}}}f_{yx}p_{x}\log\left(\frac{f_{yx}p_{x}}{q_{y}}\right).$$


**Proof of Lemma** **2.**Applying *K* to *f* yields
$$\begin{array}{cc} K(f) & =-\sum_{x,y}f_{yx}p_{x}\log\left(f_{yx}\right)-\sum_{x}p_{x}\log\left(p_{x}\right)+\sum_{y}q_{y}\log\left(q_{y}\right)\\ & =-\sum_{x,y}f_{yx}p_{x}\log\left(f_{yx}\right)-\sum_{x}\left(\sum_{y}f_{yx}\right)p_{x}\log\left(p_{x}\right)+\sum_{y}\left(\sum_{x}f_{yx}p_{x}\right)\log\left(q_{y}\right)\\ & =-\sum_{x\in X\backslash N_{p}}\sum_{y\in Y\backslash N_{f_{x}}}f_{yx}p_{x}\log\left(\frac{f_{yx}p_{x}}{q_{y}}\right).\;□ \end{array}$$

**Proof of Proposition** **2.**
(i)The non-negativity of *K* follows from Lemma 2 and the equality $q_{y}=\sum_{x^{\prime }\in X}f_{yx^{\prime }}p_{x^{\prime }}\geq f_{yx}p_{x}$.(ii)This follows from the fact that H(f|p)=0 for all deterministic *f*.(iii)Let p:•⇝X be a probability measure, and let $\left(Y_{x},q^{x}\right)\overset{Q^{x}}{⇝}\left(Y_{x}^{\prime },q^{\prime x}\right)$ be a collection of morphisms in FinPS indexed by *X*. Then the *p*-weighted convex sum ${⨁}_{x\in X}p_{x}Q^{x}$ is a morphism in FinPS of the form $\left(Z,r\right)\overset{h}{⇝}\left(Z^{\prime },r^{\prime }\right)$, where $Z:={∐}_{x\in X}Y_{x}$, $Z^{\prime }:={∐}_{x\in X}Y_{x}^{\prime }$, $h:={⨁}_{x\in X}p_{x}Q^{x}$, $r:={⨁}_{x\in X}p_{x}q^{x}$, and $r^{\prime }:={⨁}_{x\in X}p_{x}q^{\prime x}$. Then
$$\begin{array}{ccc} K(h) & = & -\sum_{z\in Z}r_{z}\log\left(r_{z}\right)+\sum_{z^{\prime }\in Z^{\prime }}r_{z^{\prime }}^{\prime }\log\left(r_{z^{\prime }}^{\prime }\right)-\sum_{z\in Z}\sum_{z^{\prime }\in Z^{\prime }}r_{z}h_{z^{\prime }z}\log\left(h_{z^{\prime }z}\right)\\ & = & -\sum_{x\in X}\sum_{y_{x}\in Y_{x}}\sum_{y_{x}^{\prime }\in Y_{x}^{\prime }}p_{x}q_{y_{x}}^{x}Q_{y_{x}y_{x}^{\prime }}^{x}\log\left(Q_{y_{x}y_{x}^{\prime }}^{x}\right)\\ & = & \sum_{x\in X}p_{x}H\left(Q^{x}|q^{x}\right), \end{array}$$
which shows that *K* is convex linear.(iv)Let $\left(X^{\left(n\right)},p^{\left(n\right)}\right)\overset{f^{\left(n\right)}}{⇝}\left(Y^{\left(n\right)},q^{\left(n\right)}\right)$ be a sequence (indexed by n∈N) of probability-preserving stochastic maps such that $X^{\left(n\right)}=X$ and $Y^{\left(n\right)}=Y$ for large enough *n*, and where $\lim_{n\to \infty }f^{\left(n\right)}=f,\lim_{n\to \infty }p^{\left(n\right)}=p,$ and $\lim_{n\to \infty }q^{\left(n\right)}=q$. Then
$$\lim_{n\to \infty }K\left(f^{\left(n\right)}\right)=-\lim_{n\to \infty }\sum_{x,y}f_{yx}^{\left(n\right)}p_{x}^{\left(n\right)}\log\left(\frac{f_{yx}^{\left(n\right)}p_{x}^{\left(n\right)}}{\sum_{x^{\prime }}f_{yx^{\prime }}^{\left(n\right)}p_{x^{\prime }}^{\left(n\right)}}\right)=K\left(f\right),$$
where the last equality follows from the fact that the limit and sum (which is finite) can be interchanged and all expressions are continuous on [0,1].(v)This follows from
$$\begin{array}{cc} H\left(\vartheta (f)\right) & =-\sum_{\begin{array}{c} x\in X\\ y\in Y \end{array}}f_{yx}p_{x}\log\left(f_{yx}p_{x}\right)\\ & =-\sum_{\begin{array}{c} x\in X\\ y\in Y \end{array}}f_{yx}p_{x}\log\left(f_{yx}\right)-\underset{=1}{\underset{︸}{\sum_{y\in Y}f_{yx}}}\sum_{x\in X}p_{x}\log\left(p_{x}\right)\\ & =H(f|p)+H(p) \end{array}$$
and the fact that $K\left(\pi_{Y}\right)=H\left(\vartheta (f)\right)-H\left(q\right)$. □


**Remark** **3.**
*Since conditional entropy vanishes for deterministic morphisms, conditional information loss restricts to FinPD as the information loss functor. It is important to note that if the term H(f|p) was not included in the expression for K(f), then the inequality K(f)≥0 would fail in general. When f is deterministic, Baez, Fritz, and Leinster proved H(p)−H(q)≥0. However, when f is stochastic, the inequality H(p)−H(q)≥0 does not hold in general. This has to do with the fact that stochastic maps may increase entropy, whereas deterministic maps always decrease it(while this claim holds in the classical setting as stated, it no longer holds for quantum systems [8]). As such, the term H(f|p) is needed to retain non-negativity as one attempts to extend BFL’s functor K on FinPD to a function on FinPS.*


Item (v) of Proposition 2 says that the conditional information loss of a map $\left(X,p\right)\overset{f}{⇝}\left(Y,q\right)$ in FinPS is the information loss of the deterministic map $\left(X\times Y,\vartheta \left(f\right)\right)\overset{\pi_{Y}}{\to }\left(Y,q\right)$ in FinPD, so that conditional information loss of a morphism in FinPS may always be reduced to the information loss of a deterministic map in FinPD naturally associated with it having the same target. This motivates the following definition.

**Definition** **16.**
*A function F:FinPS→BR is*
**reductive**
*if and only if $F\left(f\right)=F\left(\pi_{Y}\right)$ for every morphism $\left(X,p\right)\overset{f}{⇝}\left(Y,q\right)$ in FinPS (cf. Proposition 2 item (v) for notation).*


**Proposition** **3**(Reductive characterization of conditional information loss)**.**
*Let $F:\mathrm{FinPS}\to BR_{\geq 0}$ be a function satisfying the following conditions.*
*(i)* *F restricted to FinPD is functorial, convex linear, and continuous.**(ii)* *F is reductive.*
*Then F is a non-negative multiple of conditional information loss. Conversely, conditional information loss satisfies conditions (i) and (ii).*

**Proof.** This follows immediately from Theorem 1 and item (v) of Proposition 2. □

In what follows, we will characterize conditional information loss without any explicit reference to the subcatgeory FinPD or the information loss functor of Baez, Fritz, and Leinster. To do this, we first need to develop some machinery.

## 5. Coalescable Morphisms and Semi-Functoriality

While conditional information loss is not functorial on FinPS, we know it acts functorially on deterministic maps. As such, it is natural to ask for which pairs of composable stochastic maps does the conditional information loss act functorially. In this section, we answer this question, and then we use our result to define a property of functions FinPS→BR that is a weakening of functoriality, and which we refer to as *semi-functoriality*. Our definitions are valid in any Markov category (cf. Appendix B).

**Definition** **17.***A deterministic map $Z\times X\overset{h}{\to }Y$ is said to be a***mediator***for the composable pair $\left(X,p\right)\overset{f}{⇝}\left(Y,q\right)\overset{g}{⇝}\left(Z,r\right)$ in FinPS if and only if*(1)$${\left(g\circ f\right)}_{zx}=g_{zh(z,x)}f_{h(z,x)x}\;\;\mathrm{for}\;\mathrm{all}\;\left(z,x\right)\in Z\times \left(X\backslash N_{p}\right).$$*If in fact Equation (Equation 1) holds for all (z,x)∈Z×X, then h is said to be a***strong mediator***for the composable pair $X\overset{f}{⇝}Y\overset{g}{⇝}Z$ in***FinStoch**.

**Remark** **4.**
*Mediators do not exist for general composable pairs, as one can see by considering any composable pair $\left(•,1\right)\overset{p}{⇝}\left(X,p\right)\overset{f}{⇝}\left(Y,q\right)$ such that H(ϑ(f))≠H(q) (cf. Definitions 7 and 13).*


**Proposition** **4.**
*Let $\left(X,p\right)\overset{f}{⇝}\left(Y,q\right)\overset{g}{⇝}\left(Z,r\right)$ be a composable pair of morphisms in FinPS. Then the following statements are equivalent.*
*(a)* *For every $x\in X\backslash N_{p}$ and z∈Z, there exists* at most *one y∈Y such that $g_{zy}f_{yx}\neq 0$.**(b)* 
*The pair (f,g) admits a mediator $Z\times X\overset{h}{\to }Y$.*
*(c)* 
*There exists a function $Z\times X\overset{h}{\to }Y$ such that*
(2)$$h_{y(z,x)}{\left(g\circ f\right)}_{zx}p_{x}=g_{yz}f_{yx}p_{x}\;\forall \left(z,y,x\right)\in Z\times Y\times X.$$



**Proof.** ((a)⇒(b)) For every $\left(z,x\right)\in Z\times \left(X\backslash N_{p}\right)$ for which such a *y* exists, set h(z,x):=y. If no such *y* exists or if $x\in N_{p}$, set h(z,x) to be anything. Then *h* is a mediator for (f,g).((b)⇒(c)) Let *h* be a mediator for (f,g). Since (Equation 2) holds automatically for $x\in N_{p}$, suppose $x\in X\backslash N_{p}$, in which case (Equation 2) is equivalent to $h_{y(z,x)}{\left(g\circ f\right)}_{zx}=g_{yz}f_{yx}$ for all (z,y)∈Z×Y. This follows from Equation (Equation 1) and the fact that *h* is a function.((c)⇒(a)) Let $\left(z,x\right)\in Z\times \left(X\backslash N_{p}\right)$ and suppose ${\left(g\circ f\right)}_{zx}>0$. If *h* is the mediator, then $\sum_{y\in Y}g_{zy}f_{yx}={\left(g\circ f\right)}_{zx}=g_{zh(z,x)}f_{h(z,x)x}$. But since $g_{zy}f_{yx}=0$ for all y≠h(z,x), there is only one non-vanishing term in this sum, and it is precisely $g_{zh(z,x)}f_{h(z,x)x}$. □

**Theorem** **2**(Functoriality of Conditional Entropy)**.**
*Let $\left(X,p\right)\overset{f}{⇝}\left(Y,q\right)\overset{g}{⇝}\left(Z,r\right)$ be a composable pair of morphisms in FinPS. Then*
(3)H(g∘f|p)=H(f|p)+H(g|q)
*holds if and only if there exists a mediator $Z\times X\overset{h}{\to }Y$ for $\left(X,p\right)\overset{f}{⇝}\left(Y,q\right)\overset{g}{⇝}\left(Z,r\right)$.*

We first prove two lemmas.

**Lemma** **3.**
*Let $\left(X,p\right)\overset{f}{⇝}\left(Y,q\right)\overset{g}{⇝}\left(Z,r\right)$ be a pair of composable morphisms. Then*
$$H\left(\left(g\times {\mathrm{id}}_{Y}\right)\circ \Delta_{Y}\circ f|p\right)=H\left(g|q\right)+H\left(f|p\right).$$
*In particular, H(g∘f|p)=H(g|q)+H(f|p) if and only if $H\left(g\circ f|p\right)=H\left(\left(g\times {\mathrm{id}}_{Y}\right)\circ \Delta_{Y}\circ f|p\right)$.*


**Proof of Lemma** **3.**On components, ${\left(\left(g\times {\mathrm{id}}_{Y}\right)\circ \Delta_{Y}\circ f\right)}_{(z,y)x}=g_{zy}f_{yx}$. Hence,
$$\begin{array}{cc} H\left(\left(g\times {\mathrm{id}}_{Y}\right)\circ \Delta_{Y}\circ f|p\right) & =-\sum_{x}p_{x}\left(\sum_{y,z}g_{zy}f_{yx}\log\left(g_{zy}f_{yx}\right)\right)\\ & =-\sum_{x,y,z}p_{x}g_{zy}f_{yx}\log\left(g_{zy}\right)-\sum_{x,y,z}p_{x}g_{zy}f_{yx}\log\left(f_{yx}\right)\\ & =-\sum_{y,z}q_{y}g_{zy}f_{yx}\log\left(g_{zy}\right)-\sum_{x,y}p_{x}f_{yx}\log\left(f_{yx}\right)\\ & =H(g|q)+H(f|p). \end{array}$$
Note that this equality still holds if $g_{zy}=0$ or $f_{yx}=0$ as each step in this calculation accounted for such possibilities. □

**Lemma** **4.**
*Let $\left(X,p\right)\overset{f}{⇝}\left(Y,q\right)\overset{g}{⇝}\left(Z,r\right)$ be a pair of composable morphisms in FinPS. Then*
(4)$$\begin{array}{cc} 0 & \leq H\left(\left(g\times {\mathrm{id}}_{Y}\right)\circ \Delta_{Y}\circ f|p\right)-H\left(g\circ f|p\right)\\ \; & =-\sum_{x\in X\backslash N_{p}}\sum_{y\in Y\backslash N_{f_{x}}}\sum_{z\in Z\backslash N_{g_{y}}}p_{x}g_{zy}f_{yx}\log\left(\frac{g_{zy}f_{yx}}{\sum_{y^{\prime }}g_{zy^{\prime }}f_{y^{\prime }x}}\right). \end{array}$$
*Note that the order of the sums matters in this expression and also note that it is always well-defined since $g_{zy}f_{yx}\neq 0$ implies ${\left(g\circ f\right)}_{zx}\neq 0$.*


**Proof of Lemma** **4.**For convenience, temporarily set $ℵ:=H\left(\left(g\times {\mathrm{id}}_{Y}\right)\circ \Delta_{Y}\circ f|p\right)-H\left(g\circ f|p\right)$. Then
$$\begin{array}{cc} ℵ & =-\sum_{x,y,z}p_{x}g_{zy}f_{yx}\log\left(g_{zy}f_{yx}\right)+\sum_{x,y,z}p_{x}g_{zy}f_{yx}\log\left({\left(g\circ f\right)}_{zx}\right)\\ & =-\sum_{x\in X\backslash N_{p}}\sum_{y\in Y\backslash N_{f_{x}}}\sum_{z\in Z\backslash N_{g_{y}}}p_{x}g_{zy}f_{yx}\log\left(\frac{g_{zy}f_{yx}}{{\left(g\circ f\right)}_{zx}}\right), \end{array}$$
which proves the claim due to the definition of the composition of stochastic maps. □

**Proof of Theorem** **2.**Temporarily set $ℵ:=H\left(\left(g\times {\mathrm{id}}_{Y}\right)\circ \Delta_{Y}\circ f|p\right)-H\left(g\circ f|p\right)$. In addition, note that the set of all $x\in X\backslash N_{p}$ and $z\in Z\backslash N_{{\left(g\circ f\right)}_{x}}$ can be given a more explicit description in terms of the joint distribution $•\overset{s:=\gamma \circ \vartheta (g\circ f)}{⇝}Z\times X$ associated with the composite g∘f and prior *p*, namely $s_{\left(z,x\right)}:={\left(g\circ f\right)}_{zx}p_{x}$. Then,
(5)$$\left\{\left(z,x\right)\;:\;x\in X\backslash N_{p},\;z\in Z\backslash N_{{\left(g\circ f\right)}_{x}}\right\}=\left(Z\times X\right)\backslash N_{s}.$$
(⇒) Suppose ℵ=0, which is equivalent to Equation (Equation 3) by Lemma 3. Then since each term in the sum from Lemma 4 is non-negative,
$$0=-g_{zy}f_{yx}p_{x}\log\left(\frac{g_{zy}f_{yx}}{\sum_{y^{\prime }}g_{zy^{\prime }}f_{y^{\prime }x}}\right)\;\forall \;x\in X\backslash N_{p},\;y\in Y\backslash N_{f_{x}},\;z\in Z\in N_{g_{y}}.$$
Hence, fix such an $x\in X\backslash N_{p},y\in Y\backslash N_{f_{x}},z\in Z\in N_{g_{y}}$. The expression here vanishes if and only if
(6)$$g_{zy}f_{yx}={\left(g\circ f\right)}_{zx},\;i.e.,\;g_{zy^{\prime }}f_{y^{\prime }x}=0\forall y^{\prime }\in Y\backslash \left\{y\right\}.$$
Hence, for every $x\in X\backslash N_{p}$ and $z\in Z\backslash N_{{\left(g\circ f\right)}_{x}}$, there exists a unique y∈Y such that $g_{zy}f_{yx}\neq 0$. But by (Equation 5), this means that for every $\left(z,x\right)\in \left(Z\times X\right)\backslash N_{s}$, there exists a unique y∈Y such that $g_{zy}f_{yx}\neq 0$. This defines a function $\left(Z\times X\right)\backslash N_{s}\to Y\backslash N_{q}$ which can be extended in an *s*-a.e. unique manner to a function $Z\times X\overset{h}{\to }Y.$We now show the function *h* is in fact a mediator for the composable pair (g,f). The equality clearly holds if $x\in N_{p}$ since both sides vanish. Hence, suppose that $x\in X\backslash N_{p}$. Given y∈Y,z∈Z, the left-hand-side of (2) is given by
$$\delta_{yh(z,x)}{\left(g\circ f\right)}_{zx}p_{x}=\left\{\begin{array}{cc} g_{zy}f_{yx}p_{x} & \mathrm{if}\;z\in Z\backslash N_{{\left(g\circ f\right)}_{x}}\mathrm{and}\;y=h\left(z,x\right)\\ 0 & \mathrm{otherwise}. \end{array}\right.$$
by Equation (Equation 6). Similarly, if $x\in X\backslash N_{p}$ and $z\in N_{{\left(g\circ f\right)}_{x}}$, then $g_{zy}f_{yx}=0$ for all y∈Y because otherwise ${\left(g\circ f\right)}_{zx}p_{x}$ would be nonzero. If instead $z\in Z\backslash N_{{\left(g\circ f\right)}_{x}}$, then $g_{zh(z,x)}f_{h(z,x)x}\neq 0$ and $g_{zy}f_{yx}=0$ for all y∈Y\{h(z,x)} by (Equation 6). Therefore, (Equation 2) holds.(⇐) Conversely, suppose a mediator *h* exists and let $X\overset{k}{⇝}Z\times Y$ be the stochastic map given on components by $k_{(z,y)x}:=h_{y(z,x)}{\left(g\circ f\right)}_{zx}.$ Then
$$\begin{array}{cc} H(g|q)+H(f|p) & =H\left(\left(g\times {\mathrm{id}}_{Y}\right)\circ \Delta_{Y}\circ f|p\right)\;\;\mathrm{by}\;\mathrm{Lemma}\;3\\ & =H(k|p)\;\;\;\mathrm{by}\;\mathrm{Proposition}\;4\;\mathrm{item}\;\left(c\right)\\ & =-\sum_{x\in X\backslash N_{p}}\sum_{z\in Z\backslash N_{{\left(g\circ f\right)}_{x}}}\sum_{y\in Y}h_{y(z,x)}{\left(g\circ f\right)}_{zx}p_{x}\log\left(h_{y(z,x)}{\left(g\circ f\right)}_{zx}\right)\\ & =-\;\;\;\;\;\;\sum_{\left(z,x\right)\in \left(Z\times X\right)\backslash N_{s}}\sum_{y\in Y}\delta_{yh(z,x)}{\left(g\circ f\right)}_{zx}p_{x}\log\left(\delta_{yh(z,x)}{\left(g\circ f\right)}_{zx}\right)\\ & =-\;\;\;\;\sum_{\left(z,x\right)\in \left(Z\times X\right)\backslash N_{s}}{\left(g\circ f\right)}_{zx}p_{x}\log\left({\left(g\circ f\right)}_{zx}\right)\\ & =H\left((g\circ f)|p\right), \end{array}$$
as desired. □

**Corollary** **1**(Functoriality of Conditional Information Loss)**.**
*Let $\left(X,p\right)\overset{f}{⇝}\left(Y,q\right)\overset{g}{⇝}\left(Z,r\right)$ be a composable pair of morphisms in FinPS. Then K(g∘f)=K(f)+K(g) if and only if there exists a mediator $Z\times X\overset{h}{\to }Y$ for the pair $\left(X,p\right)\overset{f}{⇝}\left(Y,q\right)\overset{g}{⇝}\left(Z,r\right)$.*

**Proof.** Since the Shannon entropy difference is always functorial, the conditional information loss is functorial on a pair of morphisms if and only if the conditional entropy is functorial on that pair. Theorem 2 then completes the proof. □

**Example** **2.**
*In the notation of Theorem 2, suppose that f is*
**a.e. deterministic**
*, which means $f_{yx}=\delta_{yf(x)}$ for all $x\in X\backslash N_{p}$ for some function f (abusive notation is used). In this case, the deviation from functoriality, (4), simplifies to*
$$H\left(\left(g\times {\mathrm{id}}_{Y}\right)\circ \Delta_{Y}\circ f|p\right)-H\left(g\circ f|p\right)=-\sum_{x\in X\backslash N_{p}}\sum_{z\in Z}p_{x}g_{zf(x)}\log\left(\frac{g_{zf(x)}}{g_{zf(x)}}\right)=0.$$
*Therefore, if f is p-a.e. deterministic, H(g|q)+H(f|p)=H(g∘f|p). In this case, the mediator $Z\times X\overset{h}{\to }Y$ is given by $h:={!}_{Z}\times f$.*


**Definition** **18.**
*A pair $\left(X,p\right)\overset{f}{⇝}\left(Y,q\right)\overset{g}{⇝}\left(Z,r\right)$ of composable morphisms in FinPS is called*
**a.e. coalescable**
*if and only if $\left(X,p\right)\overset{f}{⇝}\left(Y,q\right)\overset{g}{⇝}\left(Z,r\right)$ admits a mediator $Z\times X\overset{h}{\to }Y$. Similarly, a pair $X\overset{f}{⇝}Y\overset{g}{⇝}Z$ of composable morphisms in*
**FinStoch**
*is called*
**coalescable**
*iff $X\overset{f}{⇝}Y\overset{g}{⇝}Z$ admits a strong mediator $Z\times X\overset{h}{\to }Y$.*


**Remark** **5.**
*Example 2 showed that if $\left(X,p\right)\overset{f}{⇝}\left(Y,q\right)$ is p-a.e. deterministic, then the pair $\left(X,p\right)\overset{f}{⇝}\left(Y,q\right)\overset{g}{⇝}\left(Z,r\right)$ is a.e. coalescable for any g. In particular, every pair of composable morphisms in FinPD is coalescable.*


In light of Theorem 2 and Corollary 1, we make the following definition, which will serve as one of the axioms in our later characterizions of both conditional information loss and conditional entropy.

**Definition** **19.**
*A function F:FinPS→BR is said to be*
**semi-functorial**
*iff F(g∘f)=F(g)+F(f) for every a.e. coalescable pair $\left(X,p\right)\overset{f}{⇝}\left(Y,q\right)\overset{g}{⇝}\left(Z,r\right)$ in FinPS.*


**Example** **3.**
*By Theorem 2 and Corollary 1, conditional information loss and conditional entropy are both semi-functorial.*


**Proposition** **5.**
*Suppose F:FinPS→BR is semi-functorial. Then the restriction of F to FinPD is functorial. In particular, if F is, in addition, convex linear, continuous, and reductive, then F is a non-negative multiple of conditional information loss.*


**Proof.** By Example 2, every pair of composable morphisms in FinPD is a.e. coalescable. Therefore, *F* is functorial on FinPD. The second claim then follows from Proposition 3. □

The following lemma will be used in later sections and serves to illustrate some examples of a.e. coalescable pairs.

**Lemma** **5.**
*Let $\left(W,s\right)\overset{e}{\to }\left(X,p\right)\overset{f}{⇝}\left(Y,q\right)\overset{g}{\to }\left(Z,r\right)$ be a triple of composable morphisms with e deterministic and g invertible. Then each of the following pairs are a.e. coalescable:*
*(i)* 
$\left(W,s\right)\overset{e}{\to }\left(X,p\right)\overset{f}{⇝}\left(Y,q\right)$
*(ii)* 
$\left(X,p\right)\overset{f}{⇝}\left(Y,q\right)\overset{g}{\to }\left(Z,r\right)$
*(iii)* 
$\left(W,s\right)\overset{e}{\to }\left(X,p\right)\overset{g\circ f}{⇝}\left(Z,r\right)$
*(iv)* 
$\left(W,s\right)\overset{f\circ e}{⇝}\left(Y,q\right)\overset{g}{\to }\left(Z,r\right)$



**Proof.** The proof that $\left(W,s\right)\overset{e}{\to }\left(X,p\right)\overset{f}{⇝}\left(Y,q\right)$ is coalescable was provided (in a stronger form) in Example 2. To see that $\left(X,p\right)\overset{f}{⇝}\left(Y,q\right)\overset{g}{\to }\left(Z,r\right)$ is coalescable, note that since *g* is an isomorphism we have ${\left(g\circ f\right)}_{zx}=g_{zg^{-1}\left(z\right)}f_{g^{-1}\left(z\right)x}$. Thus, $Z\times X\overset{g^{-1}\times {!}_{X}}{\to }Y\times •≅Y$ is a mediator function for g∘f, thus g∘f is coalescable. The last two claims follow from the proofs of the first two claims. □

## 6. Bayesian Inversion

In this section, we recall the concepts of a.e. equivalence and Bayesian inversion phrased in a categorical manner [2,3,9], as they will play a significant role moving forward.

**Definition** **20.**
*Let $\left(X,p\right)\overset{f}{⇝}\left(Y,q\right)$ and $\left(X,p\right)\overset{g}{⇝}\left(Y,q\right)$ be two morphisms in FinPS with the same source and target. Then f and g are said to*
**almost everywhere equivalent**
*(or p-a.e.*
**equivalent**
*) if and only if $f_{yx}=g_{yx}$ for every x∈X with $p_{x}\neq 0$. In such a case, the p-a.e. equivalence of f and g will be denoted $f\underset{p}{=}g$.*


**Theorem** **3**(Bayesian Inversion [2,9,10])**.**
*Let $\left(X,p\right)\overset{f}{⇝}\left(Y,q\right)$ be a morphism in FinPS. Then there exists a morphism $\left(Y,q\right)\overset{\bar{f}}{⇝}\left(X,p\right)$ such that ${\bar{f}}_{xy}q_{y}=f_{yx}p_{x}$ for all x∈X and y∈Y. Furthermore, for any other morphism $\left(Y,q\right)\overset{{\bar{f}}^{\prime }}{⇝}\left(X,p\right)$ satisfying this condition, $\bar{f}\underset{q}{=}{\bar{f}}^{\prime }$.*

**Definition** **21.**
*The morphism $\left(Y,q\right)\overset{\bar{f}}{⇝}\left(X,p\right)$ appearing in Theorem 3 will be referred to as a*
**Bayesian inverse**
*of $\left(X,p\right)\overset{f}{⇝}\left(Y,q\right)$. It follows that ${\bar{f}}_{xy}=p_{x}f_{yx}/q_{y}$ for all y∈Y with $q_{y}\neq 0$.*


**Proposition** **6.**
*Bayesian inversion satisfies the following properties.*
*(i)* 
*Suppose $\left(X,p\right)\overset{f}{⇝}\left(Y,q\right)$ and $\left(X,p\right)\overset{g}{⇝}\left(Y,q\right)$ are p-a.e. equivalent, and let $\bar{f}$ and $\bar{g}$ be Bayesian inverses of f and g, respectively. Then $\bar{f}\underset{q}{=}\bar{g}$.*
*(ii)* 
*Given two morphisms $\left(X,p\right)\overset{f}{⇝}\left(Y,q\right)$ and $\left(Y,q\right)\overset{g}{⇝}\left(X,p\right)$ in FinPS, then f is a Bayesian inverse of g if and only if g is a Bayesian inverse of f.*
*(iii)* 
*Let $\left(Y,q\right)\overset{\bar{f}}{⇝}\left(X,p\right)$ be a Bayesian inverse of $\left(X,p\right)\overset{f}{⇝}\left(Y,q\right)$, and let γ:X×Y→Y×X be the swap map (as in Definition 4). Then $\vartheta \left(f\right)=\gamma \circ \vartheta \left(\bar{f}\right)$*
*(iv)* 
*Let (f,g) be a composable pair of morphisms in FinPS, and suppose $\bar{f}$ and $\bar{g}$ are Bayesian inverses of f and g respectively. Then $\left(\bar{g},\bar{f}\right)$ is a composable pair, and $\bar{f}\circ \bar{g}$ is a Bayesian inverse of g∘f.*



**Proof.** These are immediate consequences of the categorical definition of a Bayesian inverse (see [3,10,11] for proofs). □

**Definition** **22.**
*A contravariant function B:FinPS→FinPS is said to be a*
**Bayesian inversion functor**
*if and only if B acts as the identity on objects and B(f) is a Bayesian inverse of f for all morphisms f in FinPS.*


This is mildly abusive terminology since functoriality only holds in the a.e. sense, as explained in the following remark.

**Remark** **6.***A Bayesian inversion functor exists. Given any $\left(X,p\right)\overset{f}{⇝}\left(Y,q\right)$, set $Y\overset{\bar{f}}{⇝}X$ to be given by ${\bar{f}}_{xy}=p_{x}f_{yx}/q_{y}$ for all y∈Y with $q_{y}\neq 0$ and ${\bar{f}}_{xy}=1/\left|X\right|$ for all y∈Y with $q_{y}=0$. Note that this does* not *define a functor. Indeed, if (X,p) is a probability space with $p_{x_{0}}=0$ for some $x_{0}\in X$, then ${\left(\bar{{\mathrm{id}}_{X}}\right)}_{x_{0}}$ is the uniform measure on X instead of the Dirac delta measure concentrated on $x_{0}$. In other words, $\bar{{\mathrm{id}}_{X}}\neq {\mathrm{id}}_{X}$. Similar issues of measure zero occur, indicating that $\bar{g\circ f}\neq \bar{f}\circ \bar{g}$ for a composable pair of morphisms $\left(X,p\right)\overset{f}{⇝}\left(Y,q\right)\overset{g}{⇝}\left(Z,r\right)$. Nevertheless, Bayesian inversion is a.e. functorial in the sense that $\bar{g\circ f}\underset{r}{=}\bar{f}\circ \bar{g}$ and $\bar{{\mathrm{id}}_{\left(X,p\right)}}\underset{p}{=}{\mathrm{id}}_{\left(X,p\right)}$.*

**Corollary** **2.**
*$B^{2}\left(f\right)\underset{p}{=}f$ for any Bayesian inversion functor B and every $\left(X,p\right)\overset{f}{⇝}\left(Y,q\right)$ in FinPS.*


**Proposition** **7.**
*Let B be a Bayesian inversion functor on FinPS (as in Definition 22). Then B is*
**a.e. convex linear**
*in the sense that*
$$B\left(\underset{x\in X}{⨁}p_{x}Q^{x}\right)\underset{q^{\prime }}{=}\underset{x\in X}{⨁}p_{x}B\left(Q^{x}\right),$$
*where $q^{\prime }:={⨁}_{x\in X}p_{x}q^{\prime x}$ and the other notation is as in Definition 8.*


**Proof.** First note that it is immediate that B is convex linear on objects since Bayesian inversion acts as the identity on objects. Let p:•⇝X be a probability measure, $\left(Y_{x},q^{x}\right)\overset{Q^{x}}{⇝}\left(Y_{x}^{\prime },q^{\prime x}\right)$ be a collection of morphisms in FinPS indexed by *X*, and suppose B is a Bayesian inversion functor. Then for $\left(z,z^{\prime }\right)\in Y_{x}\times Y_{x}^{\prime }$ with $p_{x}q_{z^{\prime }}^{\prime x}\neq 0$, we have
$$\begin{array}{ccc} B{\left(\underset{x\in X}{⨁}p_{x}Q^{x}\right)}_{zz^{\prime }} & = & \frac{p_{x}q_{z}^{x}{\left({⨁}_{x\in X}p_{x}Q^{x}\right)}_{z^{\prime }z}}{p_{x}q_{z^{\prime }}^{\prime x}}=\frac{q_{z}^{x}\left(p_{x}Q_{z^{\prime }z}^{x}\right)}{q_{z^{\prime }}^{\prime x}}=p_{x}\left(\frac{q_{z}^{x}Q_{z^{\prime }z}^{x}}{q_{z^{\prime }}^{\prime x}}\right)\\ & = & p_{x}B{\left(Q^{x}\right)}_{zz^{\prime }}={\left(\underset{x\in X}{⨁}p_{x}B\left(Q^{x}\right)\right)}_{zz^{\prime }}. \end{array}$$Thus, B is a.e. convex linear. □

**Proposition** **8.**
*Given $\left(X,p\right)\overset{f}{⇝}\left(Y,q\right)\overset{g}{⇝}\left(Z,r\right)$ in FinPS, and let $\bar{f}$ and $\bar{g}$ be Bayesian inverses of f and g respectively. Then (f,g) is a.e. coalescable if and only if $\left(\bar{g},\bar{f}\right)$ is a.e. coalescable.*


**Proof.** Since Bayesian inversion is a dagger functor on a.e. equivalence classes ([3], (Remark 13.10)), it suffices to prove one direction in this claim. Hence, suppose (f,g) is a.e. coalescable and let *h* be a mediator function realizing this. Then h∘γ is a mediator for $\left(\bar{g},\bar{f}\right)$ because
$$\begin{array}{cc} {\bar{f}}_{xy}{\bar{g}}_{yz}r_{z} & ={\bar{f}}_{xy}g_{zy}q_{y}=g_{zy}f_{yx}p_{x}=h_{y(z,x)}{\left(g\circ f\right)}_{zx}p_{x}\\ & =h_{y(z,x)}{\bar{\left(g\circ f\right)}}_{xz}r_{z}={\left(h\circ \gamma \right)}_{y(x,z)}{\bar{\left(g\circ f\right)}}_{xz}r_{z}. \end{array}$$
A completely string-diagrammatic proof is provided in Appendix B. □

The following proposition is a reformulation of the conditional entropy identity H(Y|X)+H(X)=H(X|Y)+H(Y) in terms of Bayesian inversion.

**Proposition** **9.**
*Let $\left(X,p\right)\overset{f}{⇝}\left(Y,q\right)$ be a morphism in FinPS, and suppose $\bar{f}$ is a Bayesian inverse of f. Then*
(7)$$H\left(f|p\right)+H\left(p\right)=H\left(\bar{f}|q\right)+H\left(q\right).$$


**Proof.** This follows from the fact that both sides of (Equation 7) are equal to H(ϑ(f)). □

Proposition 9 implies Bayesian inversion takes conditional entropy to conditional information loss and vice versa, which is formally stated as follows.

**Corollary** **3.**
*Let $K:\mathrm{FinPS}\to BR_{\geq 0}$ and $H:\mathrm{FinPS}\to BR_{\geq 0}$ be given by conditional information loss and conditional entropy, respectively, and let B:FinPS→FinPS be a Bayesian inversion functor. Then, H=K∘B and K=H∘B.*


**Remark** **7.**
*If $\left(X,p\right)\overset{f}{\to }\left(Y,q\right)$ is a deterministic morphism in FinPS, Baez, Fritz, and Leinster point out that the information loss of f is in fact the conditional entropy of x given y [1]. Here, we see this duality as a special case of Corollary 3 applied to deterministic morphisms.*


## 7. Bloom-Shriek Factorization

We now introduce a simple, but surprisingly useful, factorization for every morphism in FinPS, and we use it to prove some essential lemmas for our characterization theorems for conditional information loss and conditional entropy, which appear in the following sections.

**Definition** **23.**
*Given a stochastic map $X\overset{f}{⇝}Y$, the*
**bloom of *f***
*is the stochastic map $X\overset{{¡}_{f}}{⇝}X\times Y$ given by the composite $X\overset{\Delta_{X}}{⇝}X\times X\overset{{\mathrm{id}}_{X}\times f}{⇝}X\times Y$, and the*
**shriek of *f***
*is the deterministic map $X\times Y\overset{{!}_{f}}{\to }X$ given by the projection $\pi_{X}$.*


**Proposition** **10.**
*Let $\left(X,p\right)\overset{f}{⇝}\left(Y,q\right)$ be a morphism in FinPS. Then the following statement hold.*
*(i)* 
*The composite $\left(X,p\right)\overset{{¡}_{f}}{⇝}\left(X\times Y,\vartheta \left(f\right)\right)\overset{{!}_{f}}{\to }\left(X,p\right)$ is equal to the identity $id_{X}$.*
*(ii)* 
*The morphism f equals the composite $\left(X,p\right)\overset{{¡}_{f}}{⇝}\left(X\times Y,\vartheta \left(f\right)\right)\overset{{!}_{\bar{f}}\circ \gamma }{\to }\left(Y,q\right)$, where $\bar{f}$ denotes any Bayesian inverse of f and γ:X×Y→Y×X is the swap map.*
*(iii)* 
*The pair $\left(X,p\right)\overset{{¡}_{f}}{⇝}\left(X\times Y,\vartheta \left(f\right)\right)\overset{{!}_{\bar{f}}\circ \gamma \equiv \pi_{Y}}{\to }\left(Y,q\right)$ is coalescable.*



**Definition** **24.**
*The decomposition in item (ii) Proposition 10 will be referred to as the*
**bloom-shriek factorization**
*of f.*


**Proof of Propostion** **10.**Element-wise proofs are left as exercises. Appendix B contains an abstract proof using string diagrams in Markov categories. □

The bloom of *f* can be expressed as a convex combination of simpler morphisms up to isomorphism. To describe this and its behavior under convex linear semi-functors, we introduce the notion of an invariant and examine some of its properties.

**Definition** **25.**
*A function F:FinPS→BR is said to be an*
**invariant**
*if and only if for every triple of composable morphisms $\left(W,s\right)\overset{e}{\to }\left(X,p\right)\overset{f}{⇝}\left(Y,q\right)\overset{g}{\to }\left(Z,r\right)$ such that e and g are isomorphisms, then F(f)=F(g∘f∘e).*


**Lemma** **6.**
*If a function $F:\mathrm{FinPS}\to BR_{\geq 0}$ is semi-functorial, then F is an invariant.*


**Proof.** Consider a composable triple $\left(W,s\right)\overset{e}{\to }\left(X,p\right)\overset{f}{⇝}\left(Y,q\right)\overset{g}{\to }\left(Z,r\right)$ such that *e* and *g* are isomorphisms. Then
$$F\left(g\circ f\circ e\right)=F\left(g\circ (f\circ e)\right)=F\left(g\right)+F\left(f\circ e\right)=F\left(g\right)+F\left(f\right)+F\left(e\right)$$
by Lemma 5. Secondly, since *g* and *e* are isomorphisms, and since the pairs $\left(g,g^{-1}\right)$ and $\left(e,e^{-1}\right)$ are coalescable, $F\left(\mathrm{id}\right)=F\left(g\circ g^{-1}\right)=F\left(g\right)+F\left(g^{-1}\right)$. But since F(id)=0 (by semi-functoriality), this requires that F(g)=0 for an isomorphism *g* since F(g)≥0 and $F(g^{-1})\geq 0$. The same is true for F(e). Hence, $F\left(g\circ f\circ e\right)=F\left(f\right)$. □

**Lemma** **7.**
*Let $\left(X,p\right)\overset{f}{⇝}\left(Y,q\right)$ be a morphism in FinPS, and suppose $F:\mathrm{FinPS}\to BR_{\geq 0}$ is semi-functorial and convex linear. Then the following statements hold.*
*(i)* 
*$F\left(f\right)=F\left({!}_{\bar{f}}\right)+F\left({¡}_{f}\right)$*
*(ii)* 
*$F\left({¡}_{f}\right)=\sum_{x\in X}p_{x}F\left({¡}_{f_{x}}\right)$*
*(iii)* 
*$F\left({!}_{f}\right)=\sum_{x\in X}p_{x}F\left({!}_{f_{x}}\right)$*



**Proof.** For item (i), we have
$$\begin{array}{cccc} F(f) & =F({!}_{\bar{f}}\circ \gamma \circ {¡}_{f}) & \; & \;\mathrm{by}\;\mathrm{item}\;\left(\mathrm{ii}\right)\;\mathrm{of}\;\mathrm{Proposition}\;10\\ \; & =F\left({!}_{\bar{f}}\circ \gamma \right)+F\left({¡}_{f}\right) & \; & \;\mathrm{by}\;\mathrm{item}\;\left(\mathrm{iii}\right)\;\mathrm{of}\;\mathrm{Proposition}\;10\\ \; & =F\left({!}_{\bar{f}}\right)+F\left({¡}_{f}\right) & \; & \;\mathrm{by}\;\mathrm{Lemma}\;6. \end{array}$$For items (ii) and (iii), note that ${¡}_{f}$ and ${!}_{f}$ can be expressed as composites of isomorphisms and certain convex combinations, namely

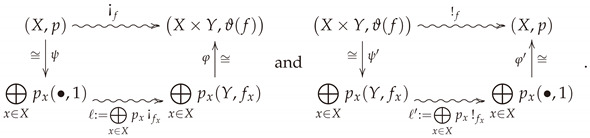
(8)
Hence,
$$\begin{array}{cccccc} F({¡}_{f}) & =F(\varphi \circ \ell \circ \psi ) & F({!}_{f}) & =F(\varphi^{\prime }\circ \ell^{\prime }\circ \psi^{\prime }) & \; & \;\mathrm{by}\;\left(8\right)\\ \; & =F(\ell ) & \; & =F(\ell^{\prime }) & \; & \;\mathrm{by}\;\mathrm{Lemma}\;6\\ \; & =\sum_{x\in X}p_{x}F\left({¡}_{f_{x}}\right) & \; & =\sum_{x\in X}p_{x}F\left({!}_{f_{x}}\right) & \; & \;\mathrm{since}\;F\;\mathrm{is}\;\mathrm{convex}\;\mathrm{linear}.\;□ \end{array}$$

**Proposition** **11.**
*Suppose $F:\mathrm{FinPS}\to BR_{\geq 0}$ is semi-functorial and convex linear. If f,g:(X,p)⇝(Y,q) are two morphisms in FinPS such that $f\underset{p}{=}g$, then F(f)=F(g).*


**Proof.** Suppose $\left(X,p\right)\overset{f}{⇝}\left(Y,q\right)$ and $\left(X,p\right)\overset{g}{⇝}\left(Y,q\right)$ are such that $f\underset{p}{=}g$, and let $\bar{f}$ and $\bar{g}$ be Bayesian inverses for *f* and *g*. Then
$$\begin{array}{cccc} F(f) & =F\left({!}_{\bar{f}}\right)+F\left({¡}_{f}\right) & \; & \;\mathrm{by}\;\mathrm{item}\;\left(i\right)\;\mathrm{of}\;\mathrm{Lemma}\;7\\ \; & =\sum_{y\in Y}q_{y}F\left({!}_{{\bar{f}}_{y}}\right)+\sum_{x\in X}p_{x}F\left({¡}_{f_{x}}\right) & \; & \;\mathrm{by}\;\mathrm{items}\;\left(\mathrm{ii}\right)\;\mathrm{and}\;\left(\mathrm{iii}\right)\;\mathrm{of}\;\mathrm{Lemma}\;7\\ \; & =\sum_{y\in Y}q_{y}F\left({!}_{{\bar{g}}_{y}}\right)+\sum_{x\in X}p_{x}F\left({¡}_{g_{x}}\right) & \; & \;\mathrm{since}\;f\underset{p}{=}g\;\mathrm{and}\;\bar{f}\underset{q}{=}\bar{g}\\ \; & =F\left({!}_{\bar{g}}\right)+F\left({¡}_{g}\right) & \; & \;\mathrm{by}\;\mathrm{items}\;\left(\mathrm{ii}\right)\;\mathrm{and}\;\left(\mathrm{iii}\right)\;\mathrm{of}\;\mathrm{Lemma}\;7\\ \; & =F(g) & \; & \;\mathrm{by}\;\mathrm{item}\;\left(i\right)\;\mathrm{of}\;\mathrm{Lemma}\;7, \end{array}$$
as desired. □

## 8. An Intrinsic Characterization of Conditional Information Loss

**Theorem** **4.**
*Suppose $F:\mathrm{FinPS}\to BR_{\geq 0}$ is a function satisfying the following conditions.*
*1.* 
*F is semi-functorial.*
*2.* 
*F is convex linear.*
*3.* 
*F is continuous.*
*4.* 
*$F({¡}_{p})=0$ for every probability distribution $•\overset{p}{⇝}X$.*

*Then F is a non-negative multiple of conditional information loss. Conversely, conditional information loss satisfies conditions 1–4.*


**Proof.** Suppose *F* satisfies conditions 1–4, let $\left(X,p\right)\overset{f}{⇝}\left(Y,q\right)$ be an arbitrary morphism in FinPS, and let γ:Y×X→X×Y be the swap map, so that ${!}_{\bar{f}}=\pi_{Y}\circ \gamma$. Then
$$\begin{array}{cccc} F(f) & =F\left({!}_{\bar{f}}\right)+F\left({¡}_{f}\right) & \; & \;\mathrm{by}\;\mathrm{item}\;\left(i\right)\;\mathrm{of}\;\mathrm{Lemma}\;7\\ \; & =F\left({!}_{\bar{f}}\right)+\sum_{x\in X}p_{x}F\left({¡}_{f_{x}}\right) & \; & \;\mathrm{by}\;\mathrm{item}\;\left(\mathrm{ii}\right)\;\mathrm{of}\;\mathrm{Lemma}\;7\\ \; & =F({!}_{\bar{f}}) & \; & \;\mathrm{by}\;\mathrm{condition}\;4\\ \; & =F\left(\pi_{Y}\circ \gamma \right)\\ \; & =F(\pi_{Y}) & \; & \;\mathrm{since}\;F\;\mathrm{is}\;\mathrm{an}\;\mathrm{invariant}\;\mathrm{by}\;\mathrm{Lemma}\;6. \end{array}$$Thus, *F* is reductive (see Definition 16) and Proposition 5 applies. □

**Remark** **8.***Under the assumption that F:FinPS→BR is semi-functorial and convex linear, one may show F satisfies condition 4 in Theorem 4 if and only if F is reductive (see Definition 16 and Proposition 5). While the reductive axiom specifies how the semi-functor acts on* all *morphisms in FinPS, condition 4 in Theorem 4 only specifies how it acts on morphisms from the* initial *object. This gives not just a simple mathematical criterion, but one with a simple intuitive interpretation as well. Namely, condition 4 says that if a process begins with no prior information, then there is no information to be lost in the process.*

We now use Theorem 4 and Bayesian inversion to prove a statement dual to Theorem 4.

**Theorem** **5.**
*Suppose $F:\mathrm{FinPS}\to BR_{\geq 0}$ is a function satisfying the following conditions.*
*1.* 
*F is semi-functorial.*
*2.* 
*F is convex linear.*
*3.* 
*F is continuous.*
*4.* 
*$F({!}_{p})=0$ for every probability distribution $•\overset{p}{⇝}X$.*


*Then F is a non-negative multiple of conditional entropy. Conversely, conditional entropy satisfies conditions 1–4.*


Before giving a proof, we introduce some terminology and prove a few lemmas. We also would like to point out that condition 4 may be given an operational interpretation as follows: if a communication channel has a constant output, then it has no conditional entropy.

**Definition** **26.**
*Let F:FinPS→BR be a function and let B be a Bayesian inversion functor. Then $\bar{F}:=F\circ B$ will be referred to as a*
**Bayesian reflection**
*of F.*


**Remark** **9.**
*By Proposition 11, if F:FinPS→BR is a convex linear semi-functor, then a Bayesian reflection is independent of the choice of a Bayesian inversion functor, and as such, is necessarily unique.*


**Lemma** **8.**
*Let $\left(X,p\right)\overset{f}{⇝}\left(Y,q\right)$ be a morphism in FinPS, suppose $F:\mathrm{FinPS}\to BR_{\geq 0}$ is a convex linear semi-functor, and let $\bar{f}$ be a Bayesian inverse of f. Then $\bar{F}\left(f\right)=F\left(\bar{f}\right)$.*


**Proof of Lemma** **8.**Let B be a Bayesian inversion functor, so that $B\left(f\right)\underset{q}{=}\bar{f}$. Then $\bar{F}\left(f\right)=F\left(B\left(f\right)\right)=F\left(\bar{f}\right),$ where the last equality follows from Proposition 11. □

**Lemma** **9.**
*Let B:FinPS→FinPS be a Bayesian inversion functor and let $\left(X_{n},p_{n}\right)\overset{f_{n}}{⇝}(Y_{n},q_{n})$ be a sequence of morphisms in FinPS converging to $\left(X,p\right)\overset{f}{⇝}\left(Y,q\right)$. Then $\lim_{n\to \infty }B\left(f_{n}\right)\underset{q}{=}B\left(f\right)$.*


**Proof of Lemma** **9.**Set $f^{\left(n\right)}:=f_{n}$. For all y∈Y with $q_{y}\neq 0$, we have
$$\lim_{n\to \infty }B{\left(f^{\left(n\right)}\right)}_{xy}=\lim_{n\to \infty }\frac{p_{x}^{\left(n\right)}f_{yx}^{\left(n\right)}}{q_{y}^{\left(n\right)}}=\frac{p_{x}f_{yx}}{q_{y}}=B{\left(f\right)}_{xy}.\;□$$

**Lemma** **10.**
*Suppose $F:\mathrm{FinPS}\to BR_{\geq 0}$ is a function satisfying conditions 1–4 of Theorem 5. Then the Bayesian reflection $\bar{F}$ is a non-negative multiple of conditional information loss.*


**Proof of Lemma** **10.**We show $\bar{F}$ satisfies conditions 1–4 of Theorem 4. Throughout the proof, let B denote a Bayesian inversion functor, so that $\bar{F}=F\circ B$.Semi-functoriality: Suppose $\left(X,p\right)\overset{f}{⇝}\left(Y,q\right)\overset{g}{⇝}\left(Z,r\right)$ is an a.e. coalescable pair of composable morphisms in FinPS. Then
$$\begin{array}{cccc} \bar{F}\left(g\circ f\right) & =F\left(\bar{g\circ f}\right) & \; & \;\mathrm{by}\;\mathrm{Lemma}\;8\;\\ \; & =F\left(\bar{f}\circ \bar{g}\right) & \; & \;\mathrm{by}\;\mathrm{item}\;\left(\mathrm{iv}\right)\;\mathrm{of}\;\mathrm{Proposition}\;6\;\\ \; & =F\left(\bar{f}\right)+F\left(\bar{g}\right) & \; & \;\mathrm{by}\;\mathrm{Proposition}\;8\\ \; & =\bar{F}\left(f\right)+\bar{F}\left(g\right) & \; & \;\mathrm{by}\;\mathrm{Lemma}\;8. \end{array}$$Thus, $\bar{F}$ is semi-functorial.Convex Linearity: Given any probability space (X,p) and a family of morphisms $\left(Y_{x},q^{x}\right)\overset{Q^{x}}{⇝}\left(Y_{x}^{\prime },q^{\prime x}\right)$ in FinPS indexed by *X*,
$$\begin{array}{cccc} \bar{F}\left(\underset{x\in X}{⨁}p_{x}Q^{x}\right) & =F\left(\underset{x\in X}{⨁}p_{x}B\left(Q^{x}\right)\right) & \; & \;\mathrm{by}\;\mathrm{Propositions}\;7\;\mathrm{and}\;11\\ \; & =\sum_{x\in X}p_{x}F\left(B(Q^{x})\right) & \; & \;\sin\mathrm{ce}\;F\;\mathrm{is}\;\mathrm{convex}\;\mathrm{linear}\\ \; & =\sum_{x\in X}\bar{F}\left(Q^{x}\right) & \; & \;\mathrm{by}\;\mathrm{definition}\;\mathrm{of}\;\bar{F}. \end{array}$$Thus, $\bar{F}$ is convex linear.Continuity: This follows from Lemma 9 and Proposition 11.$\bar{F}\left({¡}_{p}\right)=0$ for every probability distribution $•\overset{p}{⇝}X$: This follows from Lemma 8, since ${!}_{p}$ is the unique Bayesian inverse of ${¡}_{p}$. □

**Proof of Theorem** **5.**Suppose F:FinPS→BR is a function satisfying conditions 1–4 of Theorem 5, and let B be a Bayesian inversion functor. Since *F* is semi-functorial and convex linear it follows from Proposition 11 that $F=\bar{F}\circ B$, and by Lemma 10 it follows that $\bar{F}=cK$ for some non-negative constant c≥0. We then have $F=\bar{F}\circ B=cK\circ B=cH$. Thus, *F* is a non-negative multiple of conditional entropy. □

## 9. A Bayesian Characterization of Conditional Entropy

We now prove a reformulation of Theorem 5, where condition 4 is replaced by a condition that we view as an ‘entropic Bayes’ rule’.

**Definition** **27.**
*A function F:FinPS→BR satisfies an*
**entropic Bayes’ rule**
*if and only if*
$$F\left(f\right)+F\left({¡}_{p}\right)=F\left(\bar{f}\right)+F\left({¡}_{q}\right)$$
*for every morphism $\left(X,p\right)\overset{f}{⇝}\left(Y,q\right)$ in FinPS and any Bayesian inverse $\bar{f}$ of f.*


**Remark** **10.**
*The entropic Bayes’ rule is an abstraction of the conditional entropy identity 7.*


**Theorem** **6**(A Bayesian characterization of conditional entropy)**.**
*Suppose $F:\mathrm{FinPS}\to BR_{\geq 0}$ is a function satisfying the following conditions.*
*1.* *F is semi-functorial.**2.* *F is convex linear.**3.* *F is continuous.**4.* *F satisfies an entropic Bayes’ rule.*
*Then F is a non-negative multiple of conditional entropy. Conversely, conditional entropy satisfies conditions 1–4.*


**Proof.** By Theorem 5, it suffices to show $F({!}_{p})=0$ for every object (X,p) in FinPS. For this, first note that ${¡}_{1}={¡}_{1}\circ {¡}_{1}$, where (•,1) is the point-mass distribution on a single point. Since *F* is semi-functorial and ${¡}_{1}\circ {¡}_{1}$ is coalescable, we have $F\left({¡}_{1}\right)=F\left({¡}_{1}\right)+F\left({¡}_{1}\right)$, which implies $F({¡}_{1})=0$. Applying the entropic Bayes’ rule from Definition 27 to the morphism ${!}_{p}:\left(X,p\right)⇝\left(•,1\right)$ yields
$$F\left({!}_{p}\right)+F\left({¡}_{p}\right)=F\left({¡}_{p}\right)+F\left({¡}_{1}\right)\Rightarrow F\left({!}_{p}\right)=F\left({¡}_{1}\right)=0,$$
as desired. □

**Remark** **11.**
*In ([6], slide 21), Fritz asked if there is a Markov category for information theory explaining the analogy between Bayes’ rule P(A|B)P(B)=P(B|A)P(A) and the conditional entropy identity H(A|B)+H(B)=H(B|A)+H(A). In light of our work, we feel we have an adequate categorical explanation for this analogy, which we now explain.*

*Let $\left(X,p\right)\overset{f}{⇝}\left(Y,q\right)$ be an arbitrary morphism in FinPS, and suppose $F:\mathrm{FinPS}\to BR_{\geq 0}$ is semi-functorial. Then the commutative diagram (cf. Definition A4)*

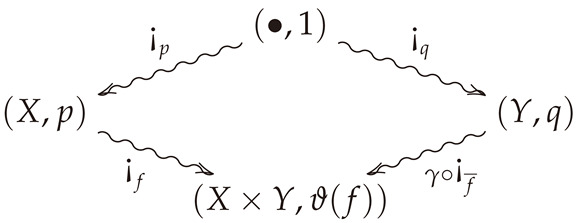
(9)
*is a coalescable square (where γ is the swap map), i.e., ${¡}_{f}\circ {¡}_{p}$ and $\left(\gamma \circ {¡}_{\bar{f}}\right)\circ {¡}_{q}$ are both coalescable. The semi-functoriality of F then implies the identity $F\left({¡}_{f}\right)+F\left({¡}_{p}\right)=F\left({¡}_{\bar{f}}\right)+F\left({¡}_{q}\right)$. Now suppose—as in the case of conditional entropy—that F satisfies the further condition that $F\left(f\right)=F\left({¡}_{f}\right)$. Then commutivity of (Equation 9) and this are equivalent to the following two respective equations:*
$$\begin{array}{rc} \underline{\mathrm{Bayes}’\;\mathrm{Rule}}: & f_{yx}p_{x}={\bar{f}}_{xy}q_{y}\\ \underline{\mathrm{Entropic}\;\mathrm{Bayes}’\;\mathrm{Rule}}: & F(f)+F\left({¡}_{p}\right)=F(\bar{f})+F\left({¡}_{q}\right). \end{array}$$
*In the case that F=H, where H is the conditional entropy, we have $H\left({¡}_{r}|1\right)=H\left(r\right)$ for every object (Z,r) in FinPS (where H(r) is the Shannon entropy). Thus, the entropic Bayes’ rule becomes $H\left(f|p\right)+H\left(p\right)=H\left(\bar{f}|q\right)+H\left(q\right)$, which is the classical identity for conditional entropy.*


## 10. Concluding Remarks

In this paper, we have provided novel characterizations of conditional entropy and the information loss of a probability-preserving stochastic map. The constructions we introduced to prove our main results are general enough to be applicable in the recent framework of synthetic probability [3]. By weakening functoriality and finding the appropriate substitute that we call *semi-functoriality*, we have shown how certain aspects of quantitative information theory can be done in this categorical framework. In particular, we have illustrated how Bayes’ rule can be formulated entropically and used as an axiom in characterizing conditional entropy.

Immediate questions from our work arise, such as the extendibility of conditional entropy to other Markov categories or even quantum Markov categories [10]. Work in this direction might offer a systematic approach towards conditional entropy in quantum mechanics. It would also be interesting to see what other quantitative features of information theory can be described from such a perspective, or if new ones will emerge.

## Figures and Tables

**Figure 1 entropy-23-01021-f001:**
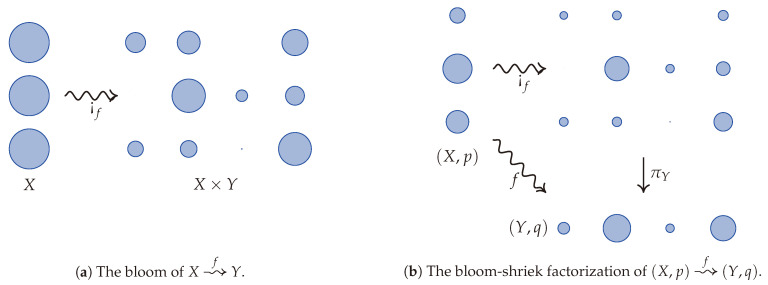
A visualization of bloom and the bloom-shriek factorization via water droplets as inspired by Gromov [5]. The bloom of *f* splits each water droplet of volume 1 (an element of *X*) into several water droplets whose total volume equates to 1. If *X* has a probability *p* on it, then the initial volume of that water droplet is scaled by this probability. The stochastic map therefore splits the water droplet using this scale.

## Data Availability

Not applicable.

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
