# Peer review of "The Information Loss of a Stochastic Map"

_entropy, 2021, doi:10.3390/e23081021_

Round 1
Reviewer 1 Report
Reviewer’s Report on entropy-1318096
The information loss of a stochastic map
By James Fullwood and Arthur J. Parzygnat
It would be better that authors can provide motivations for this work and potential real world applications.
Authors provided a stochastic extension of the Baez-Fritz-Leinster characterization of the Shannon information loss associated with a measure-preserving function and a characterization of conditional entropy in terms of the entropic Bayes’s rule that authors introduced for information measures. The work is done based on very abstract mathematics concept. Authors provided numerous theories and propositions. It is a very nice pure mathematics research. The remarks and diagrams provide nice summary of concepts. But the English writing should be polished. Specially, the abstract and introduction section. It would be better to provide concluding remarks at the end of paper.
Author Response
We thank the reviewer for their constructive comments, which we have incorporated into our manuscript. In particular, we have proofread the manuscript and polished the English accordingly, and we have added a section of concluding remarks at the end of the main body of the paper. Moreover, some potential applications have been included in Appendix A (in terms of correctable codes), and possible future directions for applications are discussed in the concluding remarks.
Reviewer 2 Report
Very good paper. The paper can be accepted as it stands now.
Author Response
We thank the reviewer for their kind words regarding our manuscript. To inform the reviewer, we added a short section of concluding remarks at the end of the main body of the paper.
Reviewer 3 Report
Authors provided a stochastic extension of the Baez-Fritz characterization of the Shannon information loss associated with a measure-preserving function. Many important theoretical propositions have been established. The explanations provided in the remarks as well as presented through diagrams are appreciated. Authors need to check possible typos and polish English writing. Specially, the writing of abstract and Introduction section.
It is also suggested that authors present concluding remarks at the end of paper.
I definitely suggest that paper be accepted subject to English writing.
Author Response

(The authors gave the same response as above.)

Round 2
Reviewer 1 Report
Adding the conclusion section makes the paper be better understood. Please remove "that" from "we introduce that ..." on line 5.
After this minor change, the paper is suggested to be accepted.